# Rapid Mining of Novel *α*-Glucosidase and Lipase Inhibitors from *Streptomyces* sp. HO1518 Using UPLC-QTOF-MS/MS

**DOI:** 10.3390/md20030189

**Published:** 2022-03-04

**Authors:** Jianlin Xu, Zhifeng Liu, Zhanguang Feng, Yuhong Ren, Haili Liu, Yong Wang

**Affiliations:** 1State Key Laboratory of Bioreactor Engineering, East China University of Science and Technology, Shanghai 200237, China; xujianlin@cemps.ac.cn (J.X.); yhren@ecust.edu.cn (Y.R.); 2CAS-Key Laboratory of Synthetic Biology, CAS Center for Excellence in Molecular Plant Sciences, Institute of Plant Physiology and Ecology, Chinese Academy of Sciences, Shanghai 200032, China; liuzhifeng@cemps.ac.cn (Z.L.); fengzhanguang@cemps.ac.cn (Z.F.); 3University of Chinese Academy of Sciences, Beijing 100039, China

**Keywords:** *Streptomyces* sp. HO1518, metabolic profiling, aminooligosaccharides, UPLC-QTOF-MS/MS, diabetes, digestive enzyme inhibitors

## Abstract

A rapid and sensitive method using ultra-high performance liquid chromatography/quadrupole time-of-flight mass spectrometry (UPLC-QTOF-MS/MS) was applied for the analysis of the metabolic profile of acarviostatin-containing aminooligosaccharides derived from *Streptomyces* sp. HO1518. A total of ninety-eight aminooligosaccharides, including eighty potential new compounds, were detected mainly based on the characteristic fragment ions originating from quinovosidic bond cleavages in their molecules. Following an LC-MS-guided separation technique, seven new aminooligosaccharides (**10**–**16**) along with four known related compounds (**17**–**20**) were obtained directly from the crude extract of strain HO1518. Compounds **10**–**13** represent the first examples of aminooligosaccharides with a rare acarviostatin II02-type structure. In addition, all isolates displayed considerable inhibitory effects on three digestive enzymes, which revealed that the number of the pseudo-trisaccharide core(s), the feasible length of the oligosaccharides, and acyl side chain exerted a crucial influence on their bioactivities. These results demonstrated that the UPLC-QTOF-MS/MS-based metabolomics approach could be applied for the rapid identification of aminooligosaccharides and other similar structures in complex samples. Furthermore, this study highlights the potential of acylated aminooligosaccharides with conspicuous *α*-glucosidase and lipase inhibition for the future development of multi-target anti-diabetic drugs.

## 1. Introduction

Type 2 diabetes mellitus (T2DM) is the most prevalent metabolic syndrome characterized by prolonged high levels of blood glucose, reflected by 537 million patients and 6.7 million deaths in 2021. The number of cases of diabetes is estimated to increase further to 783 million by 2045, which placed immense economic and social pressures on patients [1,2,3,4,5]. Currently, *α*-glucosidases (mainly *α*-amylases and disaccharidases), secreted from the intestinal chorionic epithelium capable of converting dietary carbohydrates into glucose, are still recognized as an important pharmacological target for anti-diabetic drug development. Acarbose, a typical anti-diabetes drug functioning as an *α*-glucosidases inhibitor, potently inhibits the *α*-glucosidases in vivo to retard carbohydrate digestion and avoid blood glucose elevation [6,7]. However, the specific kinase Mak1 derived from the human microbiome selectively phosphorylates acarbose at the C6-OH of C_7_N cyclohexitol ring in acarbose, leading to its inactivation [8]. Therefore, an urgent demand for the discovery of new *α*-glucosidase inhibitors with high efficacy has declared a public-health imperative for the treatment of T2DM.

Natural products, characteristic of enormous structural diversity and complexity, have been recognized as an attractive source of leading compounds and therapeutic agents attributable to their remarkable pharmacological activities [9]. Traditionally, the bioactivity-based approach remains the most commonly employed screening method to isolate natural products; however, an increasing number of compounds already described are repeatedly isolated during bioassay-guided purification [10,11]. To avoid the rediscovery of known molecules and screen new chemical entities, several dereplication strategies, including ultraviolet-visible spectroscopy (UV-Vis), nuclear magnetic resonance spectroscopy (NMR), or mass spectrometry (MS) have been developed [12,13]. Among them, MS-based dereplication has the advantage of high sensitivity and versatility, enabling users to obtain multiple types of data in a single experiment, which generally serves as the first choice of a structure-based pipeline for the discovery of unknown secondary metabolites [14]. Recently, time-of-flight mass spectrometry (TOF-MS), especially the quadrupole time-of-flight mass spectrometry (QTOF-MS), has become one of the most powerful tools for untargeted analysis of complex mixtures derived from plants and microorganisms and is attributable to their capability of providing accurate mass data and structural information [15]. Because the ultra-high performance liquid chromatography (UPLC) can shorten the analysis time of multi-component extract and increase sensitivity and reproducibility in comparison to conventional HPLC, UPLC coupled to QTOF-MS has become the crucial platform for analyzing the metabolite profiling of certain plants or microbe [16,17].

Marine *Streptomyces*, capable of producing structurally novel and biologically active secondary metabolites, has been recognized as a highly prolific resource of pharmaceutically and industrially meaningful small molecules [18,19]. In our continuing efforts to search for new anti-diabetic lead compounds from the *Streptomyces* species, we have recently discovered a series of rare acylated aminooligosaccharides with intriguing inhibitory activities against *α*-glucosidases and pancreatic lipase (PL) from the *Streptomyces* sp. HO1518 [20,21]. Structurally, these aminooligosaccharides possess a single or repeated pseudo-trisaccharide unit(s), which are combined with d-glucopyranose groups attached to the reducing and non-reducing terminus through *α*-(1→4) glycosidic bond. Pseudo-trisaccharide is composed of an acarviosine moiety and a d-glucopyranose group through *α*-(1→4) quinovosidic bond, and acarviosine is comprised of an unsaturated C_7_N cyclohexitol residue and a 4-amino-4,6-dideoxy-d-glucopyranose unit via *α*-(1→4) pseudo-glycosidic bond. Based on their structure feature, this class of naturally occurring oligosaccharide is referred to as acarviostatins followed by a Roman numeral and two digits, such as acarviostatin I01 (acarbose). The Roman numeral represents the number of the pseudo-trisaccharide cores, the middle digit denotes the number of glucose residues at the non-reducing end, and the last digit corresponds to the number of glucose units at the reducing end [22,23,24].

Since this family of oligosaccharides displayed conspicuous inhibitory activities against *α*-glucosidases and PL [20,21], this inspires our great interest to decipher the whole metabolic profiling of aminooligosaccharides in strain HO1518, which may contribute to exploring their structure-activity relationships and screening the optimal antidiabetic candidate molecules. To this end, a rapid and sensitive UPLC-QTOF-MS/MS method was developed to determine aminooligosaccharides secreted by strain HO1518 based on the MS and MS^2^ fragmentation patterns of nine reference standards (**1**–**9**) (Figure 1). This analytical approach resulted in the identification of ninety-eight oligosaccharides, including eighty new ones. Then, guided by the UPLC-QTOF-MS/MS, further study of the fermentation broth of strain HO1518 led to the isolation of seven new aminooligosaccharides (**10**–**16**) and four known congeners (**17**–**20**), among which **10**–**13** represent a new type of pseudo-octasaccharide. Compound **9** was the most potent *α*-amylase inhibitor with the IC_50_ value of 0.03 μM and was 282-fold more effective than that of acarbose (8.51 μM), while **19** exhibited the strongest activity against lipase with the IC_50_ value of 1.00 μM, and was almost equal to that of the anti-obesity orlistat (0.34 μM).

## 2. Results and Discussion

Nine reference compounds D6-*O*-acetyl-acarviostatin I03 (Ac-Aca I03, **1**), D6-*O*-propionyl-acarviostatin I03 (Pr-Aca I03, **2**), D6-*O*-isobutyryl-acarviostatin I03 (isoBu-Aca I03, **3**), D6-*O*-*β*-hydroxybutyryl-acarviostatin I03 (Hbu-Aca I03, **4**), D6-*O*-2-methyl-butyryl-acarviostatin I03 (Mbu-Aca I03, **5**), D6-*O*-isovaleryl-acarviostatin I03 (isoVa-Aca I03, **6**), D6-*O*-acetyl-acarviostatin II03 (Ac-Aca II03, **7**), D6-*O*-isobutyryl-acarviostatin II03 (isoBu-Aca II03, **8**) and acarviostatin II03 (Aca II03, **9**), previously isolated from *Streptomyces* sp. HO1518 by our group, can be grouped into two types, namely acarviostatin I03-type (**1**–**6**) and acarviostatin II03-type (**7**–**9**), based on the number of pseudo-trisaccharide units. Since the amine residues of aminooligosaccharides are readily protonated [25], the positive-ion mode HRMS/MS analysis of nine references was performed. Two series of nomenclatures bi and yj, with respect to the fragmentation of glycoconjugates in the FAB-MS/MS spectra, have been adopted in this study [26]. The bi represents fragments containing the sugar moiety counted from the non-reducing end, while the yj refers to ions possessing the aglycone at the reducing end. These fragments can provide multidimensional MS information, including retention times, molecular formulas, base peaks, sugar constituents, as well as the relative abundance of ions.

Given that **1**–**9** showed similar HRMS/MS fragmentation patterns (Appendix A), the representative D6-*O*-acetyl-acarviostatin II03 (Ac-Aca II03, **7**) harboring two pseudo-trisaccharide cores is taken as an example of how to take advantage of MS/MS data to identify the structure of acarviostatins. In the positive HRESIMS/MS spectrum of **7**, a strong protonated molecular ion at *m*/*z* 1477 was observed (Figure 2B). The high-intensity peaks in **7** were *m*/*z* 304 (b2), 769 (b5) and 1174 (y7), which resulted from the cleavages of two quinovosidic bonds. The peak intensity of some fragments with secondary amine residues, such as 146, was relatively high, which is conducive for the structure identification of aminooligosaccharides. In addition, the crucial ions at *m*/*z* 973 (b6), 1135 (b7), 1298 (b8), 854 (y5), 1012 (y6), 1174 (y7) and 1398 (y8) in **7**, were 42 mass units more than those of its deacyl product **9**, revealing that the location of the acetyl group of **7** was assigned at C-D6. Therefore, the structure of Ac-Aca II03 was established.

It is worth noting that these most abundant fragments in **7** were produced by the rupture of the quinovosidic bonds between the quinovopyranose and glucose units. Similarly, the relatively high fragment ions in the other eight standards also originated from the dissociation of the quinovosidic bond, suggesting that the cleavage of this bond was easy to achieve when compared with those of pseudo-glycosidic, glycosidic and acyl bonds (Appendix A). The resultant fragments were thus regarded as characteristic fragment ions, as outlined in Table 1. In brief, the standards **1**–**6** sharing one pseudo-trisaccharide have the same base peak at *m*/*z* 304, whereas the other references **7**–**9** possessing two pseudo-trisaccharides have the mutual fragment ions at *m*/*z* 304 and 769.

On the basis of the features of mass spectrometry data of reference, aminooligosaccharides, potential fragmentation rules of oligosaccharides were summarized. First, the glycosidic, pseudo-glycosidic and quinovosidic bonds in acarviostatins could be dissociated to some extent, and the quinovosidic bond was more fragile than two other ordinary bonds. Therefore, the most abundant signals in the positive HRMS/MS spectra were produced by the quinovosidic bond cleavages, which played pivotal roles in the structural determination of undescribed acarviostatins. Second, the fragments harboring a single or repeated amine-containing moiety (moieties) tended to display higher intensity, largely attributable to the considerably strong basicity of secondary amine residues that readily formed protonated molecules, which offered important information for the structure identification of new oligosaccharides. Third, some diagnostic product ions bi and yj in the acylated aminooligosaccharides could be applied for the assignment of the location of the acyloxy side chain.

After the establishment of the fragmentation rules of aminooligosaccharides, the UPLC-QTOF-MS/MS data of the crude extract of *Streptomyces* sp. HO1518 was analyzed. The result showed that, except for those of nine reference acarviostatins, a considerable number of newly appeared protonated molecular ions at *m*/*z* 812, 1115, 1132, 1273, 1597, 1759, etc., were detected. Further analysis of the quasi-molecular signals implied that the predicted aminooligosaccharides in strain HO1518 possess 0–3 repeating pseudo-trisaccharide moieties accompanied with a 0–1 glucose unit attached to the non-reducing end and 0–5 glucose residues on the reducing termini. In most cases, the hydroxy group at C6 of the glucose unit in pseudo-trisaccharide moiety at the proximal of the reducing terminus was acylated by an acyl group with 2–6 carbon chain. The assembly of the repeating pseudo-trisaccharide units with different numbers of glucose residues at the reducing and/or non-reducing end, together with the diversity of acyl side chain led to the identification of ninety-eight aminooligosaccharides in strain HO1518 (Appendix A), among which eighty are new compounds, including seventy-three acylated aminooligosaccharides and seven precursors. The structure of each oligosaccharide in the extract was determined on the basis of the molecular ion peak, characteristic fragments mainly corresponded to quinovosidic bond cleavages, as well as the qualitative retention time (Appendix A).

According to the abundant fragment ion peaks (*m*/*z* 304, 466, 146) arising from the cleavage of the first quinovosidic bond numbered at the non-reducing terminus, all the aminooligosaccharides are directly divided into three groups, namely acarviostatins with glucose(s) at the reducing end (Aca-glu), acarviostatins with glucose(s) at both ends (glu-Aca-glu), and acarviostatins with an incomplete pseudo-trisaccharide at the non-reducing end (incAca-glu) (Figure 3). The structures of sixty-three oligosaccharides can be categorized as group Aca-glu, which accounts for a major portion (more than 64%) of the metabolic profiling of strain HO1518. Amongst them, acarviostatin II05 (Aca II05) contains up to five glucose units at the reducing end, while D6-*O*-propionyl-acarviostatin III03 (Pr-Aca III03) possesses three pseudo-trisaccharides. The abundant MS^2^ fragment ion at *m*/*z* 304, produced by the loss of the acarviosine moiety at the non-reducing end, is the basic and characteristic peak for group Aca-glu. The structures of fifteen aminooligosaccharides are assigned as group glu-Aca-glu. Due to the presence of an additional glucose unit appended to the non-reducing terminus in group glu-Aca-glu compared with those in group Aca-glu, the typical peak ion of aminooligosaccharides is *m*/*z* 466. Group incAca-glu contains twenty aminooligosaccharides. The rare absence of an unsaturated cyclohexitol unit in the partial pseudo-trisaccharide core at the non-reducing termini results in a high-intensity characteristic MS/MS fragment at *m*/*z* 146, corresponding to the loss of the 4-amino-4,6-dideoxy-d-glucopyranose unit in the non-reducing end. Moreover, when discriminating aminooligosaccharides harbor more than one pseudo-trisaccharide core, the above-mentioned characteristic peak in each group combined with the second typical fragment ion peaks at *m*/*z* 769 for Aca-glu, 931 for glu-Aca-glu, and 611 for incAca-glu and would be greatly helpful for the judgment of their structures.

Some aminooligosaccharides, in particular the acylated acarviostatins, share identical *m*/*z* values and molecular formulas, suggesting that these compounds should be structural isomers. The reasons for this were attributed to the different assembly sequences of the same number of monosaccharides or the isomerism of the acyl side chains. For example, the protonated molecular ion at *m*/*z* 1012 in the extracted ion chromatograms (EIC) shows two peaks (Appendix A). The minor peak with a retention time of 10.19 min was assigned as D6-*O*-acetyl-acarviostatin I12 (Ac-Aca I12) attributable to the most abundant fragment at *m*/*z* 466 (Appendix A), while the major peak appearing at a retention time of 11.19 min was inferred as D6-*O*-acetyl-acarviostatin I03 (Ac-Aca I03) due to the most abundant ion at *m*/*z* 304 (Appendix A). Therefore, these abundant signals produced by the cleavage of quinovosidic bonds could confirm the number of the d-glucopyranose attached to the reducing and/or non-reducing terminus. In addition, it is worth mentioning that twenty aminooligosaccharides belonging to group incAca-glu represent a new type of oligosaccharides. To the best of our knowledge, this is the first report of acylated aminooligosaccharides directly ending with 4-amino-4-deoxy-D-quinovopyranose unit at the non-reducing end.

Motivated by the metabolic profile of aminooligosaccharides, the LC-MS guided fractionation procedure was performed to acquire new oligosaccharides. The extract of the strain HO1518 was prepared and subjected to C_18_ column chromatography to yield six fractions. After careful analysis of these fractions using LC-HRMS/MS, the molecular weights related to various aminooligosaccharides were found to enrich the fractions F1 and F2. Then, several new ion peaks at *m*/*z* 1273, 1329, 1343, 1491, and 1519 in group Aca-glu were selected as target compounds (Appendix A). In addition, many newly appeared quasi-molecular ion peaks of acarviostatins, especially those in group incAca-glu, also inspired our great interest to further perform the chemical search of strain HO1518 for the discovery of new anti-diabetic agents. Nevertheless, to our regret, we failed to acquire these novel compounds due to the trace amount of these two groups of oligomers.

Under the guidance of the aforementioned quasi-molecular peaks, seven new aminooligosaccharide congeners (**10**–**16**) and four known related compounds (**17**–**20**) were isolated (Figure 4). Compound **10**, white amorphous powder, was assigned a HRESIMS ion peak at *m*/*z* 1273.4938 ([M + H]^+^, calcd for 1273.4927), which matched a molecular formula of C_50_H_84_N_2_O_35_ with 10 degrees of hydrogen deficiency (Appendix A). The 1D and 2D NMR spectra, especially the 1D-selective TOCSY, 2D-TOCSY, HSQC, HMBC, and HSQC-TOCSY, allowed the construction of the gross structure of **10** (Figure 5). This deduction was further supported by several crucial fragments at *m*/*z* 304 (b2), 769 (b5), and 970 (y6) observed in the HRESIMS/MS spectrum of **10**, corresponding to the fission of the quinovosidic bond (Figure 6). The molecular formula of **11** was determined as C_53_H_88_N_2_O_36_ (*m*/*z* 1329.5234 ([M + H]^+^, calcd for 1329.5190) by HRESIMS, suggesting that **11** was an acetylated derivative of **10**. A careful analysis of ^1^H and ^13^C NMR spectra between **10** and **11** revealed that the hydroxyl group at C-D6 was acetylated in **11**, which was further verified by the ^1^H-^1^H COSY cross peak of H-2′/H-3′ and the HMBC correlations from H-2′ (δ_H_ 2.48) and H-3′ (δ_H_ 1.13) to C-1′ (δ_C_ 180.8) as well as H-3′ to C-2′ (δ_C_ 30.6) (Figure 5). Similarly, the carbon signals (δ_C_ 180.1, 33.9, 18.3 and 18.2 for **12**, and δ_C_ 176.2, 42.9, 25.5, 21.7 and 21.7 for **13**) indicated **12** and **13** to be isobutyryl and isovaleryl substituted analogs of **10**, respectively.

The molecular formula of **14** was assigned as C_59_H_98_N_2_O_41_ by HRESIMS ion peak at *m*/*z* 1491.5715 ([M + H]^+^, calcd for 1491.5718), which was 162 mass units more than that of **11**. Detailed inspection of the NMR spectroscopic data of **11** and **14** suggested that they should feature an analogous planar structure (Figure 4), with the major difference being the presence of an additional glucose unit at the reducing end in **14** (δ_C_ 102.3, 79.9, 76.1, 74.3, 74.1 and 63.5). After a careful comparison of ^1^H and ^13^C NMR data of **15** and **16** (Appendix A) with those of **14**, three compounds were found to share the identical acarviostatin II03-type core skeleton, but the side chain differentiated. The propionyl functionality in **14** was replaced by the butyryl group in **15** (δ_H_ 2.45, 1.65, 1.18 and 0.94) or the 2-methyl-butyryl group in **16** (δ_H_ 2.52, 1.63, 1.51, 1.13 and 0.88), which could be supported by the ^1^H-^1^H COSY cross-peaks (H-2′/H-3′/H-4′ in **15**, and H-5′/H-2′/H-3′/H-4′ in **16**) and the HRMS/MS spectra (Figure 5 and Appendix A).

T2DM is one of the most serious chronic diseases worldwide, which is closely linked to disturbances of glucose and lipid metabolism. Inhibiting *α*-glucosidase and lipase involved in the breakdown of carbohydrates and fats can reduce glucose and free fatty acid absorption in the gastrointestinal tract, which contributes to avoiding postprandial hyperglycemia and restoring normal levels of insulin secretion of pancreatic *β*-cells in diabetic patients [27,28,29]. Thus, compounds **9**–**20** were evaluated for their inhibitory activities against PPA, sucrase and PL, as presented in Table 2.

All the compounds showed inhibition of three metabolic enzymes under the assay conditions. Twelve isolates (**9**–**20**) showed remarkable inhibition of PPA, with the IC_50_ values ranging from 0.03 to 0.40 μM (Appendix A). Compounds **9**–**16**, **19** and **20** possessing the repeated pseudo-trisaccharide moieties exhibited more potential *α*-amylase inhibitory effects than those with a single pseudo-trisaccharide (**17** and **18**), among which **9** (IC_50_ = 0.03 μM) was 282-fold more effective than that of acarbose (IC_50_ = 8.51 μM). Nevertheless, the more pseudo-trisaccharide cores might pose an unfavorable effect for sucrase inhibition by comparison with the suppressing activities of **9**, **10**, and **17**. When the hydroxyl group occurred at C-6 in **10** (IC_50_ = 13.05 μM) was acylated, **11**–**13** presented stronger inhibitory activity against sucrase with IC_50_ values of 4.34, 6.79 and 7.06 μM, respectively, which implied that the acyl group shows a positive contribution to their inhibitory potency toward sucrase. Furthermore, twelve compounds displayed considerable inhibitory ability against PL with IC_50_ values in a range of 1.00–31.56 μM, while acarbose was inactive with an IC_50_ value of 191.00 μM. Of these compounds, **13**, **16**, **19** and **20**, sharing an acarviostatin II03-type structure, showed potent inhibitory effects against lipase with IC_50_ values of 1.56, 1.34, 1.00 and 1.43 μM, respectively, which was nearly equal to the positive control orlistat (IC_50_ = 0.34 μM). It is noteworthy, that increasing the acyl chain length in aminooligosaccharides contributes to enhancing their lipase inhibitory activities, as referred to **9**–**20**. These biological results highlight the potential of acylated aminooligosaccharides with prominent *α*-glucosidase and lipase inhibition for the future development of multi-target anti-diabetic drugs.

The significant PL inhibitory activity for aminooligosaccharides prompted us to further investigate the potential molecular recognition mechanism between this class of compounds and PL, thus the molecular docking simulations were implemented using a previously reported crystal structure of human PL (PDB ID: 1LPB) [30,31]. To better understand the binding mode of different aminooligosaccharides, the anti-obesity drug orlistat was firstly docked into the same domain for comparative purposes. As shown in Figure 7A, orlistat abrogated the activity of human PL by occupying the substrate binding canyon of PL and stabilized itself via strong interactions with a series of key residues (G76, F77, D79, S152 and R256) in the catalytic active site. Based on this, three characteristic aminooligosaccharides **9** (IC_50_ = 7.64 μM), **10** (IC_50_ = 12.66 μM), **17** (IC_50_ = 31.56 μM) as well as acarbose (negative control) belonging to acarviostatins II03, II02, I03 and I01 type, respectively, were selected as ligands for further detailed study.

As we anticipated, the cyclohexitol ring of the pseudo-trisaccharide unit for **9**, **10** and **17** could be perfectly docked into the catalytic cavity of PL by forming three hydrogen bonds with G76, F77 and S152, whereas the pseudo-tetrasaccharide acarbose, only located outside the catalytic pocket with no bonding site observed in the catalytic active center, indicates that the chain length and the pseudo-trisaccharide core(s) of aminooligosaccharides were crucial in hinting PL activity (Figure 7B–E). Besides the common residues mentioned above, **17** also interacted with four residues E233, C237, K238, and C261, while **10** showed interactions with the five residues E233, Q244, T255, D257, and F258. Notably, compound **9** (calculated binding energy = −7.8 kcal/mol), possessing an extra seven polar contacts with PL, exhibits a higher potency toward lipase than **10** and **17** (calculated binding energy = −7.3 and −6.8 kcal/mol), suggesting that the total strength of individual contact between ligand and PL is a definitive factor to forming a stable substrate-enzyme complex. These results were consistent with the aforementioned biological results and demonstrated the pseudo-trisaccharide unit(s) along with the glucose residues of aminooligosaccharides played a crucial role in their lipase inhibitory activity.

## 3. Materials and Methods

### 3.1. General Experimental Procedures

The optical rotations were performed on an Anton Paar MCP-500 spectropolarimeter (Anton Paar, Graz, Austria) at 20 °C. UV spectra were recorded on a JASCO V-550 UV/VIS spectrophotometer (Jasco Corporation, Tokyo, Japan). IR data were measured using a FT-IR Vertex 70 v spectrometer (Bruker, Fällanden, Switzerland). The 1D and 2D NMR spectra were acquired using a Bruker Avance 500 MHz spectrometer with TMS as an internal standard (Bruker, Fällanden, Switzerland). HRESIMS data were collected on a Thermo Q Exactive high-resolution mass spectrometer (Thermo Fisher Scientific, Waltham, MA, USA). HRMS/MS data were recorded on an Agilent Q-TOF 6545 mass spectrometer (Agilent Technologies, Santa Clara, CA, USA) equipped with an electrospray ionization source (ESI). MCI gel CHP20/P120 (Mitsubishi Chemical Corporation, Tokyo, Japan) and SiliaSphere C_18_ (50 µm, Silicycle, QuébecK, QC, Canada) were used for column chromatography. UPLC analysis was performed using an Agilent 1200Series LC system (Agilent Technologies, Santa Clara, CA, USA) equipped with a binary pump, an online degasser, an autoplate-sampler, and a thermostatically controlled column compartment. The Thermo ultimate 3000 (Thermo Fisher Scientific, Waltham, MA, USA), equipped with an Alltech 3300 ELSD detector and VWD detector was used for HPLC. Preparative HPLC was performed using a SilGreen C_18_ column (250 × 20 mm, 5 μm, 12 nm, Greenherbs CO., Ltd., Beijing, China), while semi-preparative HPLC was performed utilizing a TSK-gel 100 V C_18_ column (250 × 10 mm, 5 μm, 12 nm, Tosoh Corporation, Tokyo, Japan).

### 3.2. Reagents

The HPLC-grade methanol and acetonitrile were purchased from CINC High Purity Solvents (Shanghai) Co., Ltd. (Shanghai, China). The other solvents were of analytical grade (Sinopharm Chemical Reagent Co., Ltd., Beijing, China). Porcine pancreatic *α*-amylase (PPA) and PL were purchased from Sigma Aldrich Co. (St Louis, MO, USA). Sucrase and acarbose were obtained from Shanghai yuanye Bio-Technology Co., Ltd. (Shanghai, China). Orlistat was bought from Shanghai xushuo Bio-Technology Co., Ltd. (Shanghai, China). Standards **1**–**9** were previously isolated from *Streptomyces* sp. HO1518 by our group.

### 3.3. Bacterial Material

The *Streptomyces* sp. HO1518 was isolated from a sediment sample collected from the Rizhao coastal area, Shandong Province of China, in summer 2010. This strain (Voucher Specimen No. M2018176) is preserved at the China Center for Type Culture Collection (CCTCC), Wuhan University.

### 3.4. UPLC Analysis

The ethanol extract of strain HO1518 was separated using an XBridge C_18_ column (4.6 × 150 mm, 3.5 μm; Waters, Milford, MA, USA). The mobile phase was composed of A (0.01% aqueous ammonia) and B (acetonitrile) with a flow rate of 0.3 mL/min. The column temperature was maintained at 40 °C, and the injection volume was 1 μL. The elution program was as follows: 0–1 min, 5% B; 1–21 min, 5–35% B; 21–26 min, 35–65% B; 26–27 min, 65–100% B; 27–28.5 min, 100% B; 28.5–29 min, 100–5% B; 29–30 min, 5% B.

### 3.5. QTOF-MS/MS Analysis

HRMS/MS spectra were performed on an Agilent Q-TOF 6545 mass spectrometer equipped with an electrospray ionization source (ESI). The operating parameters were set as follows: drying gas (nitrogen, N_2_) flow rate, 6.0 L/min; drying gas temperature, 320 °C; nebulizer, 45 psig; sheath gas temperature, 350 °C; sheath gas flow, 12 L/min; capillary, 3500 V; skimmer, 65 V; OCT RF V, 750 V; and fragmentor voltage, 180 V. For MS/MS experiments, the collision energy was adjusted from 10V to 45V to optimize signals and obtain maximal structural information from the ions of interest. The system was operated under the Masshunter workstation software, version B.02.00 (Agilent Technologies, Santa Clara, CA, USA). Each sample was analyzed in positive-ion mode to provide sufficient information for structural identification. The mass range was set at *m*/*z* 50–2000.

### 3.6. Fermentation, Extraction and Isolation

The 70 L fermented broth of strain HO1518 was filtered to remove mycelia and the supernatant was subjected to XAD-16 resin by column chromatography, eluting with anhydrous ethanol to obtain the crude extract. The ethanol extract (9.2 g) was separated into 6 fractions (Frs. 1–6) on a C_18_ reverse-phase (RP) silica gel column by step gradient elution with MeOH/H_2_O (5–100%, *v*/*v*). Since the majority of aminooligosaccharide derivatives were present in Frs. 1 and 2, the MS-guided fractionation was carried out.

Fr. 1 (4.2 g) was subjected to reversed-phase C_18_ silica gel using the gradient elution with MeOH/H_2_O (5–100%, *v*/*v*) to obtain six subfractions (Frs. 1-1–1-6). Fr. 1-1 (0.2 g) was chromatographed over the MCI column and further purified by a preparative RP HPLC system equipped with a preparative SilGreen C_18_ column (MeCN/H_2_O, 8 mL/min, 6:94) to produce **10** (1.6 mg, t*_R_* 9.6 min). Fr. 1-3 (0.3 g) was fractionated by HPLC using an isocratic mobile phase of 10% MeCN/H_2_O to obtain **19** (4.5 mg, t*_R_* 15.1 min), whereas Fr. 1-4 (0.4 g) was successively separated by MCI and HPLC with an isocratic mobile phase of 10% MeCN/H_2_O to acquire **14** (13.2 mg, t*_R_* 40.5 min). Fr. 1-5 (0.8 g) was repeatedly purified by HPLC (MeCN/H_2_O, 8 mL/min, 16:84) to yield **18** (10.2 mg, t*_R_* 14.3 min). Fr. 1-6 (0.8 g) was fractionated by the MCI column, which was purified by HPLC on a preparative SilGreen C_18_ column (MeCN/H_2_O, 8 mL/min, 18:82) to yield **16** (4.2 mg, t*_R_* 12.2 min) and **20** (6.2 mg, t*_R_* 12.8 min). Fr. 2 (3.9 g) was subjected to the MCI column, eluting with MeOH/H_2_O (5–100%, *v*/*v*) to afford four subfractions (Frs. 2-1–2-4). Fr. 2-1 (0.2 g) was purified by HPLC with an isocratic phase of 18% MeOH/H_2_O to yield **17**. Fr. 2-2 (0.5 g) was separated by the reversed-phase C_18_ silica gel column, and then purified by HPLC (MeCN/H_2_O, 8 mL/min, 16:84) to obtain **12** (1.9 mg, t*_R_* 10.4 min) and **15** (5.9 mg, t*_R_* 11.9 min). Fr. 2-3 (0.3 g) was purified by HPLC using 10% MeCN/H_2_O to produce **11** (2.2 mg, t*_R_* 17.0 min), while Fr. 2-4 (0.2 g) was separated by the RP HPLC system (MeCN/H_2_O, 8 mL/min, 6:94) to afford **13** (2.0 mg, t*_R_* 44.3 min).

Acarviostatin II02 (Aca II02, **10**): White amorphous powder, ${\left[\alpha \right]}_{D}^{25}$ +127.2 (*c* 1.05, H_2_O). UV (H_2_O) end absorption; IR *ν*_max_ 3319, 1663, 1396, 1149, 1031 cm^−1^. ^1^H (500 MHz) and ^13^C (125 MHz) NMR spectroscopic data, see Appendix A; positive ESIMS: *m*/*z* 1273 [M + H]^+^; HRESIMS: *m*/*z* 1273.4938 [M + H]^+^ (calcd for C_50_H_84_N_2_O_35_, 1273.4927).

D6-*O*-Propionyl-acarviostatin II02 (Pr-Aca II02, **11**): White amorphous powder, ${\left[\alpha \right]}_{D}^{25}$+142.4 (*c* 1.01, H_2_O). UV (H_2_O) end absorption; IR *ν*_max_ 3337, 1726, 1407, 1149, 1026 cm^−1^. ^1^H (500 MHz) and ^13^C (125 MHz) NMR spectroscopic data, see Appendix A; positive ESIMS: *m*/*z* 1329 [M + H]^+^; HRESIMS: *m*/*z* 1329.5234 [M + H]^+^ (calcd for C_53_H_88_N_2_O_36_, 1329.5190).

D6-*O*-Isobutyryl-acarviostatin II02 (isoBu-Aca II02, **12**): White amorphous powder, ${\left[\alpha \right]}_{D}^{25}$ +136.1 (*c* 1.02, H_2_O). UV (H_2_O) end absorption; IR *ν*_max_ 3329, 1721, 1365, 1147, 1014 cm^−1^. ^1^H (500 MHz) and ^13^C (125 MHz) NMR spectroscopic data, see Appendix A; positive ESIMS: *m*/*z* 1343 [M + H]^+^; HRESIMS: *m*/*z* 1343.5365 [M + H]^+^ (calcd for C_54_H_90_N_2_O_36_, 1343.5346).

D6-*O*-Isovaleryl-acarviostatin II02 (isoVa-Aca II02, **13**): White amorphous powder, ${\left[\alpha \right]}_{D}^{25}$ +143.0 (*c* 0.99, H_2_O). UV (H_2_O) end absorption; IR *ν*_max_ 3305, 1647, 1407, 1150, 1013 cm^−1^. ^1^H (500 MHz) and ^13^C (125 MHz) NMR spectroscopic data, see Appendix A; positive ESIMS: *m*/*z* 1357 [M + H]^+^; HRESIMS: *m*/*z* 1357.5520 [M + H]^+^ (calcd for C_55_H_92_N_2_O_36_, 1357.5503).

D6-*O*-Propionyl-acarviostatin II03 (Pr-Aca II03, **14**): White amorphous powder, ${\left[\alpha \right]}_{D}^{25}$ +136.3 (*c* 0.90, H_2_O). UV (H_2_O) end absorption; IR *ν*_max_ 3304, 1734, 1402, 1148, 1023 cm^−1^. ^1^H (500 MHz) and ^13^C (125 MHz) NMR spectroscopic data, see Appendix A; positive ESIMS: *m*/*z* 1491 [M + H]^+^; HRESIMS: *m*/*z* 1491.5727 [M + H]^+^ (calcd for C_59_H_98_N_2_O_41_, 1491.5718).

D6-*O*-Butyryl-acarviostatin II03 (Bu-Aca II03, **15**): White amorphous powder, ${\left[\alpha \right]}_{D}^{25}$ +150.3 (*c* 0.53, H_2_O). UV (H_2_O) end absorption; IR *ν*_max_ 3324, 1729, 1568, 1149, 1024 cm^−1^. ^1^H (500 MHz) and ^13^C (125 MHz) NMR spectroscopic data, see Appendix A; positive ESIMS: *m*/*z* 1505 [M + H]^+^; HRESIMS: *m*/*z* 1505.5872 [M + H]^+^ (calcd for C_59_H_98_N_2_O_41_, 1505.5874).

D6-*O*-2-Methyl-butyryl-acarviostatin II03 (Mbu-Aca II03, **16**): White amorphous powder, ${\left[\alpha \right]}_{D}^{25}$ +130.1 (*c* 1.03, H_2_O). UV (H_2_O) end absorption; IR *ν*_max_ 3303, 1645, 1406, 1149, 1014 cm^−1^. ^1^H (500 MHz) and ^13^C (125 MHz) NMR spectroscopic data, see Appendix A; positive ESIMS: *m*/*z* 1519 [M + H]^+^; HRESIMS: *m*/*z* 1519.6039 [M + H]^+^ (calcd for C_61_H_102_N_2_O_41_, 1519.6031).

### 3.7. PPA Inhibition Assay

The PPA inhibitory activities of compounds **9**–**20** were conducted based on a previously reported method. Commercial *α*-amylase inhibitor acarbose was used as the positive control [20].

### 3.8. Sucrase Inhibition Assay

The sucrase inhibition assay of compounds **9**–**20** was evaluated according to the previously reported method. Acarbose was also used as the positive control [21].

### 3.9. PL Inhibition Assay

The lipase inhibition assay of compounds **9**–**20** was performed according to the method outlined by McDougall et al. with slight modification [32]. Orlistat was measured as a positive control.

### 3.10. Molecular Docking

The molecular docking simulations were performed by AutoDock Vina software, version 1.5.7 [33]. The crystal structure of human PL was downloaded from the RSCB Protein Data Bank (PDB ID: 1LPB). The binding site parameters (*x*: 4.342 Å; *y*: 24.299 Å; *z*: 47.471 Å) and the grid box dimensions (30 × 30 × 30 Å) were set. The results of molecular docking were evaluated on the basis of the binding energy, criteria of binding structure, and possible interactions between ligand and the critical catalytic triad of protein 1LPB.

## 4. Conclusions

In summary, the hyphenated system UPLC-QTOF-MS/MS that could provide qualitative retention time and reliable mass spectrometry information was utilized to reveal the metabolic profiling of aminooligosaccharides secreted by *Streptomyces* sp. HO1518. A total of ninety-eight aminooligosaccharides, including eighty new compounds, were detected and characterized from the extract of stain HO1518. Among them, twenty structural intriguing oligomers that ended with the 4-amino-4-deoxy-D-quinovopyranose unit at the non-reducing terminus were reported for the first time. The subsequent MS-guided fractionation method resulted in the isolation of seven new oligosaccharides (**10**–**16**) and four known analogs (**17**–**20**). Notably, compounds **10**–**13** are the first reported examples of oligosaccharides with a rarely occurring acarviostatin II02-type structure. All the compounds exhibited significant inhibitory activities against three digestive enzymes, among which compounds **9**–**16**, **19** and **20** sharing two pseudo-trisaccharides were the most effective inhibitors of *α*-amylase and lipase. Furthermore, primary structure-activity relationships of **9**–**20** revealed that the number of the pseudo-trisaccharide core and acyl side chain play pivotal roles in their biological activity, which was evidenced by molecular docking analysis. These results of this study highlighted the advantages of UPLC-QTOF-MS/MS for the rapid structural identification of oligosaccharides, and this strategy could be extended to other investigations for high-throughput analysis of natural products with similar structures. More importantly, this study not only provided new lead compounds for further scientific research towards anti-diabetic drug discovery, but also shed light on the structural optimization of aminooligosaccharides analogs for medicinal scientists.

## Figures and Tables

**Figure 1 marinedrugs-20-00189-f001:**
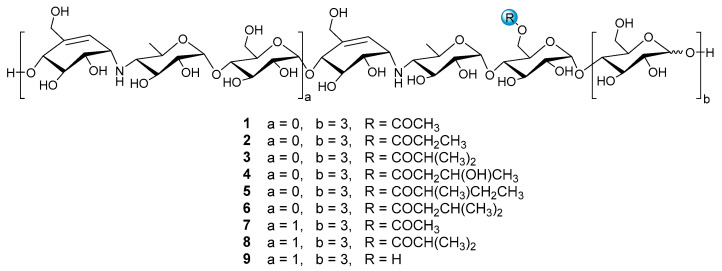
The structures of compounds **1**–**9**.

**Figure 2 marinedrugs-20-00189-f002:**
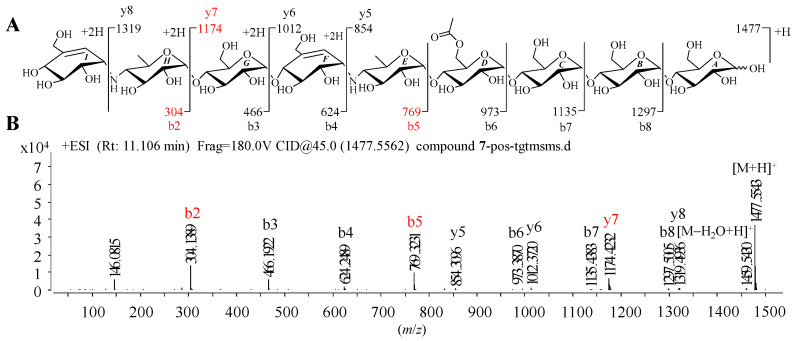
Positive HRESIMS/MS fragmentation and spectrum of **7**. (**A**) Positive-ion HRESIMS/MS fragmentation pattern of **7**; (**B**) HRESIMS/MS spectra of **7**.

**Figure 3 marinedrugs-20-00189-f003:**
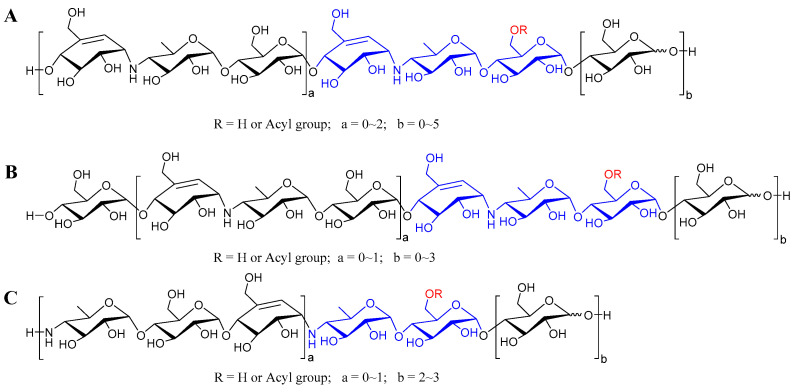
The general structures of aminooligosaccharides from *Streptomyces* sp. HO1518. (**A**) The general structures of acarviostatins with glucoses at the reducing terminus; (**B**) The general structures of acarviostatins with glucoses at the reducing and non-reducing terminus; (**C**) The general structures of acarviostatins with an incomplete pseudo-trisaccharide at the non-reducing terminus.

**Figure 4 marinedrugs-20-00189-f004:**
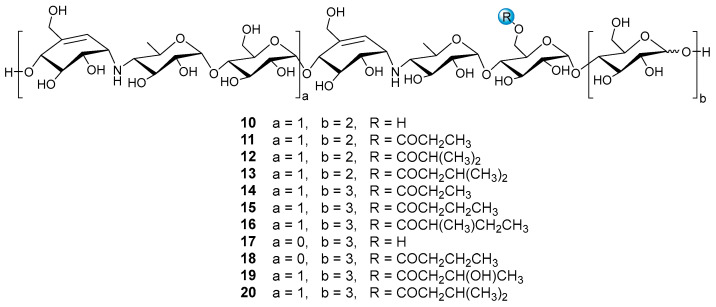
The structures of compounds **10**–**20**.

**Figure 5 marinedrugs-20-00189-f005:**
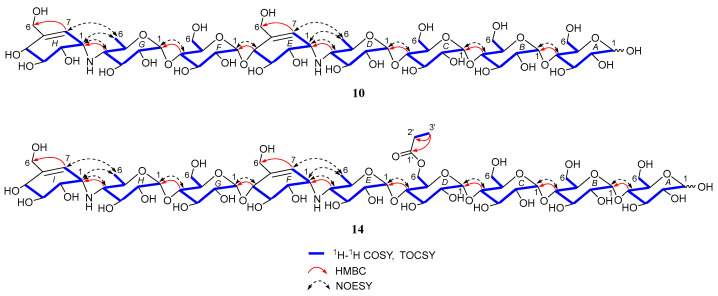
Key 2D NMR correlations of compounds **10** and **14**.

**Figure 6 marinedrugs-20-00189-f006:**
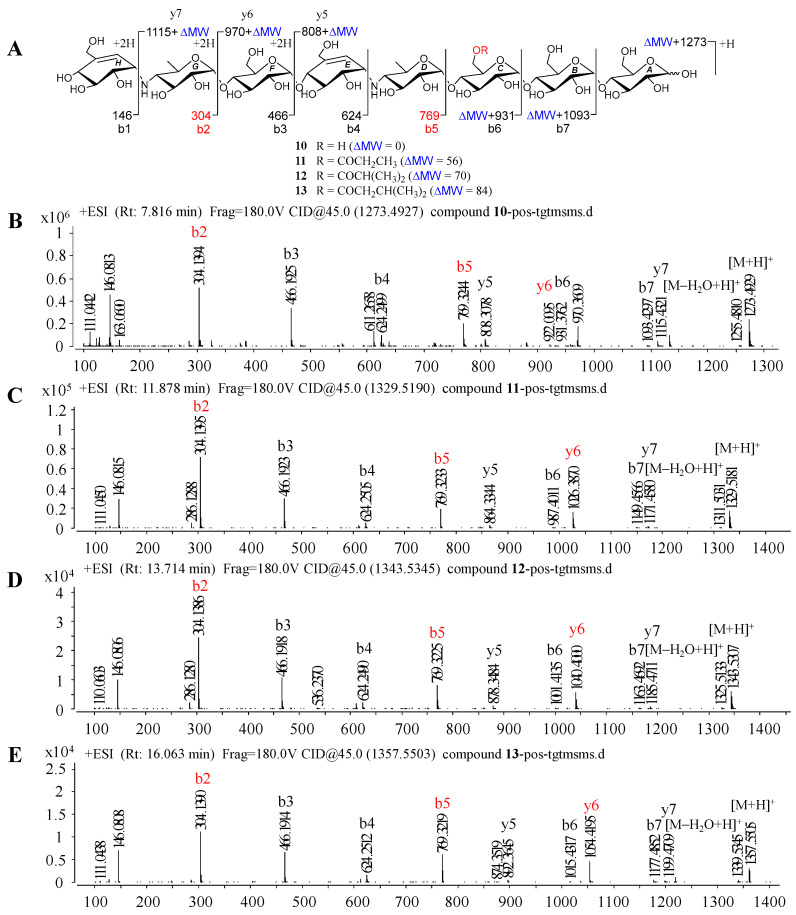
Positive HRESIMS/MS fragmentation and spectra of **10**–**13**. (**A**) Positive-ion HRESIMS/MS fragmentation patterns of **10**–**13**; (**B**–**E**) HRESIMS/MS spectra of **10**–**13**.

**Figure 7 marinedrugs-20-00189-f007:**
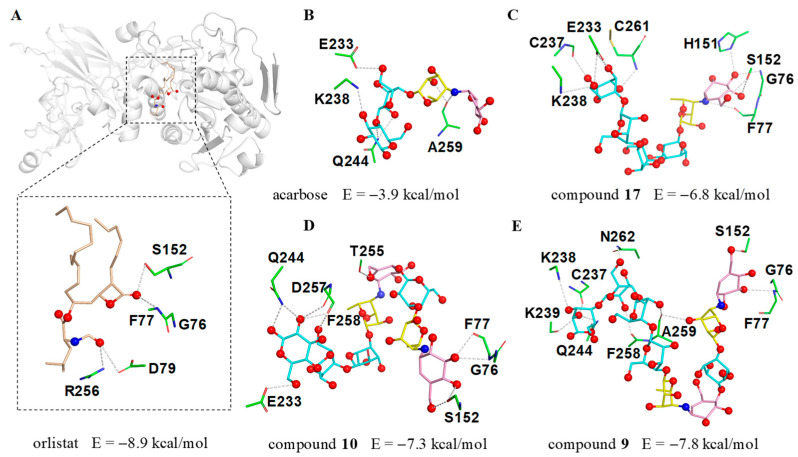
The docking results of human PL (PDB ID: 1LPB) with inhibitors. (**A**) The possible interactions between 1LPB and orlistat; (**B**) acarbose; (**C**) acarviostatin I03; (**D**) acarviostatin II02; (**E**) acarviostatin II03. Wheat: orlistat; pink: C_7_N cyclohexitol; yellow: 4-amino-4,6-dideoxy-d-glucopyranose; cyan: d-glucopyranose.

**Table 1 marinedrugs-20-00189-t001:** Information of reference aminooligosaccharides **1**–**9**.

Compounds	Formula	t*_R_* (min)	[M + H]^+^	Characteristic Fragment Ions
**1**	C_39_H_65_NO_29_	10.99	1012.3715	304.1395, 1012.3703
**2**	C_40_H_67_NO_29_	13.10	1026.3872	304.1395, 1026.3862
**3**	C_41_H_69_NO_29_	15.43	1040.4028	304.1481, 1040.4034
**4**	C_41_H_69_NO_30_	11.31	1056.3977	304.1386, 1056.3943
**5**	C_42_H_71_NO_29_	17.93	1054.4184	304.1388, 1054.4172
**6**	C_42_H_71_NO_29_	18.04	1054.4184	304.1388, 1054.4200
**7**	C_58_H_96_N_2_O_41_	11.11	1477.5561	304.1389, 769.3231, 1174.4232, 1477.5543
**8**	C_60_H_100_N_2_O_41_	14.40	1505.5874	304.1391, 769.3228, 1202.4547, 1505.5877
**9**	C_56_H_94_N_2_O_40_	9.27	1435.5456	304.1386, 769.3219, 1132.4094, 1435.5427

**Table 2 marinedrugs-20-00189-t002:** The inhibitory activities of **9**–**20** against PPA, sucrase and PL.

Compounds	IC_50_ Values (μM) ^a^		
	Against PPA	Against Sucrase	Against PL
**9**	0.030 ± 0.001	17.24 ± 0.76	7.64 ± 0.13
**10**	0.084 ± 0.001	13.05 ± 0.55	12.66 ± 0.76
**11**	0.079 ± 0.001	4.34 ± 0.24	7.22 ± 0.10
**12**	0.085 ± 0.006	6.79 ± 0.06	5.48 ± 0.18
**13**	0.092 ± 0.001	7.06 ± 0.09	1.56 ± 0.04
**14**	0.035 ± 0.001	2.56 ± 0.12	4.21 ± 0.03
**15**	0.059 ± 0.007	10.67 ± 2.60	4.46 ± 0.14
**16**	0.052 ± 0.003	7.28 ± 0.10	1.34 ± 0.03
**17**	0.296 ± 0.007	11.12 ± 0.24	31.56 ± 4.13
**18**	0.402 ± 0.008	3.80 ± 0.78	11.68 ± 2.52
**19**	0.061 ± 0.005	9.67 ± 0.10	1.00 ± 0.12
**20**	0.080 ± 0.003	9.93 ± 0.50	1.43 ± 0.08
acarbose	8.513 ± 0.240	2.34 ± 0.23	191.00 ± 15.17
orlistat	-	-	0.34 ± 0.06

^a^ Values are expressed as the mean ± SD.

## Data Availability

Not applicable.

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
