# Peer review of "Rapid Mining of Novel α-Glucosidase and Lipase Inhibitors from Streptomyces sp. HO1518 Using UPLC-QTOF-MS/MS"

_marinedrugs, 2022, doi:10.3390/md20030189_

Round 1

Reviewer 1 Report

Xu et al. conducted research in the isolation and mining of aminoligosaccharides with a UPLC-QTOF method. The manuscript descripted detailed MS-based strategy for structure dereplication and comprehensive data analysis, including MS and NMR. Bioactivity of several isolated aminooligosaccharides were also investigated, in inhibiting T2DM related enzymes. Overall, this manuscript is highly-recommended for publication. 

Suggestions:

Line 321-327. When comparing the binding potency of compound 9 and compound 10/17, it's recommended to provide a quantitative score from Autodock vina. Granted that compound 9 formed more polar contacts with PL, the total strength of individual contact was more important. Providing quantitative values would be helpful to compare with the order of their IC50s. 

Author Response

Dear reviewer,

    Many thanks for your precious suggestions/comments concerning my manuscript entitled “Rapid mining of novel α-glucosidase and lipase inhibitors from Streptomyces sp. HO1518 using UPLC-QTOF-MS/MS” (Manuscript ID: marinedrugs-1589284). We have, in connection with your concerns, modified our paper carefully. In particular, to help reader understand the structural identification of seven new compounds, the significant 2D NMR correlations (HMBC, COSY, TOCSY) of compounds 10-16 were provided in the revised supporting information (Tables S3-S9). In addition, we have deliberately proofread the manuscript to minimize typographical and grammatical errors. Now, I would like to report you, point by point, the modifications we made as follows in connection with your concerns:

Comments and Suggestions for Authors

    Xu et al. conducted research in the isolation and mining of aminoligosaccharides with a UPLC-QTOF method. The manuscript descripted detailed MS-based strategy for structure dereplication and comprehensive data analysis, including MS and NMR. Bioactivity of several isolated aminooligosaccharides were also investigated, in inhibiting T2DM related enzymes. Overall, this manuscript is highly-recommended for publication.

Suggestions:

Line 321-327. When comparing the binding potency of compound 9 and compounds 10/17, it's recommended to provide a quantitative score from Autodock vina. Granted that compound 9 formed more polar contacts with PL, the total strength of individual contact was more important. Providing quantitative values would be helpful to compare with the order of their IC50s.

Reply: We are highly appreciated for your informative suggestion! Indeed, providing quantitative values would be helpful to compare with the order of their IC50s. Following your valuable suggestion, we calculated the total binding energies of individual contact between ligand and PL of compounds 9 (-7.8 kcal/mol), 10 (-7.3 kcal/mol) and 17 (-6.8 kcal/mol) as well as orlistat (-8.9 kcal/mol) and acarbose (-3.9 kcal/mol), respectively. The quantitative values obtained are in accordance with the order of their IC50s and have been added in the revised manuscript (main text and Figure 7).

    Herewith please find the uploaded revised version (marinedrugs-1589284.revised) and the ‘track changes’ version (marinedrugs-1589284.marked). I’m looking forward to hearing from you for the further comments regarding our revised manuscript.

Best regards,

Sincerely yours,

Dr. Yong Wang

Shanghai, Feb 18, 2022

Reviewer 2 Report

The manuscript described a UPLC-QTOF-MS/MS characterisation of aminooligosaccharides derived from Streptomyces sp. HO1518, where seven new compounds were isolated and characterised by NMR. This is the third installments of their investigation into the aminooligosaccharides produced by the same organism (the previous two papers were also published in Marine Drugs, in 2018 and 2020). The techniques used (MS/MS fragmentations) to solve the structure have been mentioned in their two previous paper, so it is not a new method. The new technique featured in this manuscript is the use of UPLC-QTOF to reveal the presence of other aminooligosaccharides in the extracts.

The structural elucidations of the new compounds were sound, although a tabulated data of all 2D NMR correlations (HSQC, HMBC, COSY, TOCSY) observed will be better than the 2D spectra, which were not annotated, making it harder to see any significant correlations. Also the authors drawn all the COSY and HMBC correlations for the sugars moieties despite the highly overlapped NMR signals of the molecules. Without seeing the expanded regions of the spectra, I am not sure if they can observe all of these correlations. 

There are some uncommon expression in English which could be improved, but they don't detract from the meaning. Please do a thorough spell check and proof-read for the correct English grammar.

In the SI, some figure captions were labeled as 2D-TCOSY (should it be 2D-TOCSY) and HSQC-TCOSY (should it be HSQC-TOCSY). Please check and correct them.

I recommended that paper to be accepted but  would like to caution the authors not to fragment their publications any further. They should published all the new compounds at the same time they were isolated and not to divide them in several instalment to get more paper. 

Author Response

Dear reviewer,

    Many thanks for your precious suggestions/comments concerning my manuscript entitled “Rapid mining of novel α-glucosidase and lipase inhibitors from Streptomyces sp. HO1518 using UPLC-QTOF-MS/MS” (Manuscript ID: marinedrugs-1589284). We have, in connection with your concerns, modified our paper carefully. In particular, to help reader understand the structural identification of seven new compounds, the significant 2D NMR correlations (HMBC, COSY, TOCSY) of compounds 10-16 were provided in the revised supporting information (Tables S3-S9). In addition, we have deliberately proofread the manuscript to minimize typographical and grammatical errors. Now, I would like to report you, point by point, the modifications we made as follows in connection with your concerns:

Comments and Suggestions for Authors

    The manuscript described a UPLC-QTOF-MS/MS characterisation of aminooligosaccharides derived from Streptomyces sp. HO1518, where seven new compounds were isolated and characterised by NMR. This is the third installments of their investigation into the aminooligosaccharides produced by the same organism (the previous two papers were also published in Marine Drugs, in 2018 and 2020). The techniques used (MS/MS fragmentations) to solve the structure have been mentioned in their two previous paper, so it is not a new method. The new technique featured in this manuscript is the use of UPLC-QTOF to reveal the presence of other aminooligosaccharides in the extracts.

  1. The structural elucidations of the new compounds were sound, although a tabulated data of all 2D NMR correlations (HSQC, HMBC, COSY, TOCSY) observed will be better than the 2D spectra, which were not annotated, making it harder to see any significant correlations. Also the authors drawn all the COSY and HMBC correlations for the sugars moieties despite the highly overlapped NMR signals of the molecules. Without seeing the expanded regions of the spectra, I am not sure if they can observe all of these correlations.

Reply: We highly appreciated for your constructively valuable advice. As you know, the details structural elucidation of new aminooligosaccharides (10-16) is a tough task owing to the high overlap of 1H and 13C signals in their 1D and 2D NMR spectra caused by the complex skeleton. Therefore, 1D and 2D TOCSY were simultaneously applied to address the issues. Herein, to help better understanding for the readers, we have supplemented the crucial 2D NMR correlations (HMBC, COSY, TOCSY) of compounds 10-16 in the revised supporting information (Tables S3-S9).

  1. There are some uncommon expression in English which could be improved, but they don't detract from the meaning. Please do a thorough spell check and proof-read for the correct English grammar.

Reply: Many thanks for your constructive advice. we have deliberately proofread the manuscript and the corrected some uncommon expression.

  1. In the SI, some figure captions were labeled as 2D-TCOSY (should it be 2D-TOCSY) and HSQC-TCOSY (should it be HSQC-TOCSY). Please check and correct them.

Reply: We are sorry for our carelessness, and we have corrected these mistakes in our revised supporting information.

    I recommended that paper to be accepted but would like to caution the authors not to fragment their publications any further. They should published all the new compounds at the same time they were isolated and not to divide them in several instalment to get more paper.

    Herewith please find the uploaded revised version (marinedrugs-1589284.revised) and the ‘track changes’ version (marinedrugs-1589284.marked). I’m looking forward to hearing from you for the further comments regarding our revised manuscript.

Best regards,

Sincerely yours,

Dr. Yong Wang

Shanghai, Feb 18, 2022

Reviewer 3 Report

This is important work that would be a welcome addition to marine drugs. My major recommended change is in regard to how the NMR data is presented. I recommend, in addition to table S1, included separate tables for each compound 10, 11, 12, 13, 14, 15, 16. along with a chemdraw structure with the positions identified. I also recommend including a column with the number of hydrogens for each entry.

Author Response

Dear reviewer,

    Many thanks for your precious suggestions/comments concerning my manuscript entitled “Rapid mining of novel α-glucosidase and lipase inhibitors from Streptomyces sp. HO1518 using UPLC-QTOF-MS/MS” (Manuscript ID: marinedrugs-1589284). We have, in connection with your concerns, modified our paper carefully. In particular, to help reader understand the structural identification of seven new compounds, the significant 2D NMR correlations (HMBC, COSY, TOCSY) of compounds 10-16 were provided in the revised supporting information (Tables S3-S9). In addition, we have deliberately proofread the manuscript to minimize typographical and grammatical errors. Now, I would like to report you, point by point, the modifications we made as follows in connection with your concerns:

Comments and Suggestions for Authors

  1. This is important work that would be a welcome addition to marine drugs. My major recommended change is in regard to how the NMR data is presented. I recommend, in addition to table S1, included separate tables for each compound 10, 11, 12, 13, 14, 15, 16. along with a chemdraw structure with the positions identified.

Reply: Thanks very much for your informative comment. We have added these NMR data of compounds 10-16 in the revised supporting information (Tables S3-S9) according to your valuable suggestion

  1. I also recommend including a column with the number of hydrogens for each entry.

Reply: Many thanks for your informative suggestion. We have added a column with the number of hydrogens in the revised Tables S1-S2 in the supporting information.

    Herewith please find the uploaded revised version (marinedrugs-1589284.revised) and the ‘track changes’ version (marinedrugs-1589284.marked). I’m looking forward to hearing from you for the further comments regarding our revised manuscript.

Best regards,

Sincerely yours,

Dr. Yong Wang

Shanghai, Feb 18, 2022